# Flexible Transfer Learning under Support and Model Shift

**Xuezhi Wang**
Computer Science Department
Carnegie Mellon University
xuezhiw@cs.cmu.edu

**Jeff Schneider**
Robotics Institute
Carnegie Mellon University
schneide@cs.cmu.edu

## Abstract

Transfer learning algorithms are used when one has sufficient training data for one supervised learning task (the source/training domain) but only very limited training data for a second task (the target/test domain) that is similar but not identical to the first. Previous work on transfer learning has focused on relatively restricted settings, where specific parts of the model are considered to be carried over between tasks. Recent work on covariate shift focuses on matching the marginal distributions on observations $X$ across domains. Similarly, work on target/conditional shift focuses on matching marginal distributions on labels $Y$ and adjusting conditional distributions $P(X|Y)$, such that $P(X)$ can be matched across domains. However, covariate shift assumes that the support of test $P(X)$ is contained in the support of training $P(X)$, i.e., the training set is richer than the test set. Target/conditional shift makes a similar assumption for $P(Y)$. Moreover, not much work on transfer learning has considered the case when a few labels in the test domain are available. Also little work has been done when all marginal and conditional distributions are allowed to change while the changes are smooth. In this paper, we consider a general case where both the support and the model change across domains. We transform both $X$ and $Y$ by a location-scale shift to achieve transfer between tasks. Since we allow more flexible transformations, the proposed method yields better results on both synthetic data and real-world data.

## 1 Introduction

In a classical transfer learning setting, we have sufficient fully labeled data from the source domain (or the training domain) where we fully observe the data points $X^{tr}$, and all corresponding labels $Y^{tr}$ are known. On the other hand, we are given data points, $X^{te}$, from the target domain (or the test domain), but few or none of the corresponding labels, $Y^{te}$, are given. The source and the target domains are related but not identical, thus the joint distributions, $P(X^{tr}, Y^{tr})$ and $P(X^{te}, Y^{te})$, are different across the two domains. Without any transfer learning, a statistical model learned from the source domain does not directly apply to the target domain. The use of transfer learning algorithms minimizes, or reduces the labeling work needed in the target domain. It learns and transfers a model based on the labeled data from the source domain and the data with few or no labels from the target domain, and should perform well on the unlabeled data in the target domain. Some real-world applications of transfer learning include adapting a classification model that is trained on some products to help learn classification models for other products [17], and learning a model on the medical data for one disease and transferring it to another disease.

The real-world application we consider is an autonomous agriculture application where we want to manage the growth of grapes in a vineyard [3]. Recently, robots have been developed to take images of the crop throughout the growing season. When the product is weighed at harvest at the end of each season, the yield for each vine will be known. The measured yield can be used to learn a model

to predict yield from images. Farmers would like to know their yield early in the season so they can make better decisions on selling the produce or nurturing the growth. Acquiring training labels early in the season is very expensive because it requires a human to go out and manually estimate the yield. Ideally, we can apply a transfer-learning model which learns from previous years and/or on other grape varieties to minimize this manual yield estimation. Furthermore, if we decide that some of the vines have to be assessed manually to learn the model shift, a simultaneously applied active learning algorithm will tell us which vines should be measured manually such that the labeling cost is minimized. Finally, there are two different objectives of interest. To better nurture the growth we need an accurate estimate of the current yield of each vine. However, to make informed decisions about pre-selling an appropriate amount of the crops, only an estimate of the sum of the vine yields is needed. We call these problems active learning and active surveying respectively and they lead to different selection criteria.

In this paper, we focus our attention on real-valued regression problems. We propose a transfer learning algorithm that allows both the support on $X$ and $Y$, and the model $P(Y|X)$ to change across the source and target domains. We assume only that the change is smooth as a function of $X$. In this way, more flexible transformations are allowed than mean-centering and variance-scaling. Specifically, we build a Gaussian Process to model the prediction on the transformed $X$, then the prediction is matched with a few observed labels $Y$ (also properly transformed) available in the target domain such that both transformations on $X$ and on $Y$ can be learned. The GP-based approach naturally lends itself to the active learning setting where we can sequentially choose query points from the target dataset. Its final predictive covariance, which combines the uncertainty in the transfer function and the uncertainty in the target label prediction, can be plugged into various GP based active query selection criteria. In this paper we consider (1) Active Learning which reduces total predictive covariance [18, 19]; and (2) Active Surveying [20, 21] which uses an estimation objective that is the sum of all the labels in the test set.

As an illustration, we show a toy problem in Fig. 1. As we can see, the support of $P(X)$ in the training domain (red stars) and the support of $P(X)$ in the test domain (blue line) do not overlap, neither do the support of $Y$ across the two domains. The goal is to learn a model on the training data, with a few labeled test data (the filled blue circles), such that we can successfully recover the target function (the blue line). In Fig. 3, we show two real-world grape image datasets. The goal is to transfer the model learned from one kind of grape dataset to another. In Fig. 2, we show the labels (the yield) of each grape image dataset, along with the 3rd dimension of its feature space. We can see that the real-world problem is quite similar to the toy problem, which indicates that the algorithm we propose in this paper will be both useful and practical for real applications.

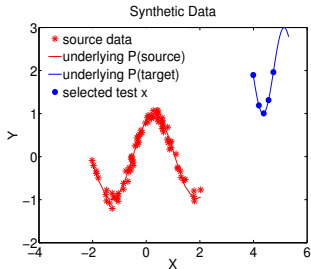

Figure 1: Toy problem

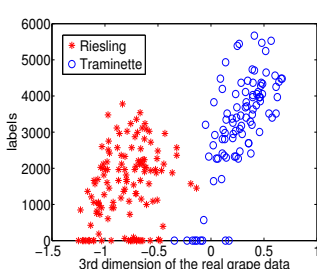

Figure 2: Real grape data

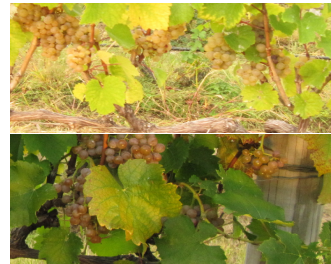

Figure 3: A part of one image from each grape dataset

We evaluate our methods on synthetic data and real-world grape image data. The experimental results show that our transfer learning algorithms significantly outperform existing methods with few labeled target data points.

## 2   Related Work

Transfer learning is applied when joint distributions differ across source and target domains. Traditional methods for transfer learning use Markov logic networks [4], parameter learning [5, 6], and Bayesian Network structure learning [7], where specific parts of the model are considered to be carried over between tasks.

Recently, a large part of transfer learning work has focused on the problem of covariate shift [8, 9, 10]. They consider the case where only $P(X)$ differs across domains, while the conditional distribution $P(Y|X)$ stays the same. The kernel mean matching (KMM) method [9, 10], is one of the algorithms that deal with covariate shift. It minimizes $||\mu(P_{te}) - \mathbf{E}_{x \sim P_{tr}(x)}[\beta(x)\phi(x)]||$ over a re-weighting vector $\beta$ on training data points such that $P(X)$ are matched across domains. However, this work suffers two major problems. First, the conditional distribution $P(Y|X)$ is assumed to be the same, which might not be true under many real-world cases. The algorithm we propose will allow more than just the marginal on $X$ to shift. Second, the KMM method requires that the support of $P(X^{te})$ is contained in the support of $P(X^{tr})$, i.e., the training set is richer than the test set. This is not necessarily true in many real cases either. Consider the task of transferring yield prediction using images taken from different vineyards. If the images are taken from different grape varieties or during different times of the year, the texture/color could be very different across transferring tasks. In these cases one might mean-center (and possibly also variance-scale) the data to ensure that the support of $P(X^{te})$ is contained in (or at least largely overlapped with) $P(X^{tr})$. In this paper, we provide an alternative way to solve the support shift problem that allows more flexible transformations than mean-centering and variance-scaling.

Some more recent research [12] has focused on modeling target shift ($P(Y)$ changes), conditional shift ($P(X|Y)$ changes), and a combination of both. The assumption on target shift is that $X$ depends causally on $Y$, thus $P(Y)$ can be re-weighted to match the distributions on $X$ across domains. In conditional shift, the authors apply a location-scale transformation on $P(X|Y)$ to match $P(X)$. However, the authors still assume that the support of $P(Y^{te})$ is contained in the support of $P(Y^{tr})$. In addition, they do not assume they can obtain additional labels, $Y^{te}$, from the target domain, and thus make no use of the labels $Y^{te}$, even if some are available.

There also have been a few papers handling differences in $P(Y|X)$. [13] designed specific methods (change of representation, adaptation through prior, and instance pruning) to solve the label adaptation problem. [14] relaxed the requirement that the training and testing examples be drawn from the same source distribution in the context of logistic regression. Similar to work on covariate shift, [15] weighted the samples from the source domain to deal with domain adaptation. These settings are relatively restricted while we consider a more general case that both the data points $X$ and the corresponding labels $Y$ can be transformed smoothly across domains. Hence all data will be used without any pruning or weighting, with the advantage that the part of source data which does not help prediction in the target domain will automatically be corrected via the transformation model.

The idea of combining transfer learning and active learning has also been studied recently. Both [22] and [23] perform transfer and active learning in multiple stages. The first work uses the source data without any domain adaptation. The second work performs domain adaptation at the beginning without further refinement. [24] and [25] consider active learning under covariate shift and still assume $P(Y|X)$ stays the same. In [16], the authors propose a combined active transfer learning algorithm to handle the general case where $P(Y|X)$ changes smoothly across domains. However, the authors still apply covariate shift algorithms to solve the problem that $P(X)$ might differ across domains, which follows the assumption covariate shift made on the support of $P(X)$. In this paper, we propose an algorithm that allows more flexible transformations (location-scale transform on both $X$ and $Y$). Our experiments on real-data shows this additional flexibility pays off in real applications.

## 3 Approach

### 3.1 Problem Formulation

We are given a set of $n$ labeled training data points, $(X^{tr}, Y^{tr})$, from the source domain where each $X_i^{tr} \in \Re^{d_x}$ and each $Y_i^{tr} \in \Re^{d_y}$. We are also given a set of $m$ test data points, $X^{te}$, from the target domain. Some of these will have corresponding labels, $Y^{teL}$. When necessary we will separately denote the subset of $X^{te}$ that has labels as $X^{teL}$, and the subset that does not as $X^{teU}$. For simplicity we restrict $Y$ to be univariate in this paper, but the algorithm we proposed easily extends to the multivariate case.

For *static transfer learning*, the goal is to learn a predictive model using all the given data that minimizes the squared prediction error on the test data, $\Sigma_{i=1}^{m}(\hat{Y}_i^{te} - Y_i^{te})^2$ where $\hat{Y}_i$ and $Y_i$ are the predicted and true labels for the $i$th test data point. We will evaluate the transfer learning algorithms

by including a subset of labeled test data chosen uniformly at random. For *active transfer learning* the performance metric is the same. The difference is that the active learning algorithm chooses the test points for labeling rather than being given a randomly chosen set.

## 3.2 Transfer Learning

Our strategy is to simultaneously learn a nonlinear mapping $X^{te} \to X^{new}$ and $Y^{te} \to Y^*$. This allows flexible transformations on both $X$ and $Y$, and our smoothness assumption using GP prior makes the estimation stable. We call this method Support and Model Shift (SMS).

We apply the following steps ($K$ in the following represents the Gaussian kernel, and $K_{XY}$ represents the kernel between matrices $X$ and $Y$, $\lambda$ ensures invertible kernel matrix):

- Transform $X^{teL}$ to $X^{new(L)}$ by a location-scale shift: $X^{new(L)} = \mathbf{W}^{teL} \odot X^{teL} + \mathbf{B}^{teL}$, such that the support of $P(X^{new(L)})$ is contained in the support of $P(X^{tr})$;

- Build a Gaussian Process on $(X^{tr}, Y^{tr})$ and predict on $X^{new(L)}$ to get $Y^{new(L)}$;

- Transform $Y^{teL}$ to $Y^*$ by a location-scale shift: $Y^* = \mathbf{w}^{teL} \odot Y^{teL} + \mathbf{b}^{teL}$, then we optimize the following empirical loss:

$$\arg\min_{\mathbf{W}^{teL}, \mathbf{B}^{teL}, \mathbf{w}^{teL}, \mathbf{b}^{teL}, \mathbf{w}^{te}} ||Y^* - Y^{new(L)}||^2 + \lambda_{reg}||\mathbf{w}^{te} - \mathbf{1}||^2, \qquad (1)$$

where $\mathbf{W}^{teL}, \mathbf{B}^{teL}$ are matrices with the same size as $X^{teL}$. $\mathbf{w}^{teL}, \mathbf{b}^{teL}$ are vectors with the same size as $Y^{teL}$ ($l$ by 1, where $l$ is the number of labeled samples in the target domain), and $\mathbf{w}^{te}$ is an $m$ by 1 scale vector on all $Y^{te}$. $\lambda_{reg}$ is a regularization parameter.

To ensure the smoothness of the transformation w.r.t. $X$, we parameterize $\mathbf{W}^{teL}, \mathbf{B}^{teL}, \mathbf{w}^{teL}, \mathbf{b}^{teL}$ using: $\mathbf{W}^{teL} = R^{teL}\mathbf{G}, \mathbf{B}^{teL} = R^{teL}\mathbf{H}, \mathbf{w}^{teL} = R^{teL}\mathbf{g}, \mathbf{b}^{teL} = R^{teL}\mathbf{h}$, where $R^{teL} = L^{teL}(L^{teL} + \lambda I)^{-1}, L^{teL} = K_{X^{teL}X^{teL}}$. Following the same smoothness constraint we also have: $\mathbf{w}^{te} = R^{te}\mathbf{g}$, where $R^{te} = K_{X^{te}X^{teL}}(L^{teL} + \lambda I)^{-1}$. This parametrization results in the new objective function:

$$\arg\min_{G, H, g, h} ||(R^{teL}\mathbf{g} \odot Y^{teL} + R^{teL}\mathbf{h}) - Y^{new(L)}||^2 + \lambda_{reg}||R^{te}\mathbf{g} - \mathbf{1}||^2. \qquad (2)$$

In the objective function, although we minimize the discrepancy between the transformed labels and the predicted labels for only the labeled points in the test domain, we put a regularization term on the transformation for all $X^{te}$ to ensure overall smoothness in the test domain. Note that the non-linearity of the transformation makes the SMS approach capable of recovering a fairly wide set of changes, including non-monotonic ones. However, because of the smoothness constraint imposed on the location-scale transformation, it might not recover some extreme cases where the scale or location change is non-smooth/discontinuous. However, under these cases the learning problem by itself would be very challenging.

We use a Metropolis-Hasting algorithm to optimize the objective (Eq. 2) which is multi-modal due to the use of the Gaussian kernel. The proposal distribution is given by $\theta^t \sim \mathcal{N}(\theta^{t-1}, \Sigma)$, where $\Sigma$ is a diagonal matrix with diagonal elements determined by the magnitude of $\theta \in \{\mathbf{G}, \mathbf{H}, \mathbf{g}, \mathbf{h}\}$. In addition, the transformation on $X$ requires that the support of $P(X^{new})$ is contained in the support of $P(X^{tr})$, which might be hard to achieve on real data, especially when $X$ has a high-dimensional feature space. To ensure that the training data can be better utilized, we relax the support-containing condition by enforcing an overlapping ratio between the transformed $X^{new}$ and $X^{tr}$, i.e., we reject those proposal distributions which do not lead to a transformation that exceeds this ratio.

After obtaining $\mathbf{G}, \mathbf{H}, \mathbf{g}, \mathbf{h}$, we make predictions on $X^{teU}$ by:

- Transform $X^{teU}$ to $X^{new(U)}$ with the optimized $\mathbf{G}, \mathbf{H}$: $X^{new(U)} = \mathbf{W}^{teU} \odot X^{teU} + \mathbf{B}^{teU} = R^{teU}\mathbf{G} \odot X^{teU} + R^{teU}\mathbf{H}$;

- Build a Gaussian Process on $(X^{tr}, Y^{tr})$ and predict on $X^{new(U)}$ to get $Y^{new(U)}$;

- Predict using optimized $\mathbf{g}, \mathbf{h}$: $\hat{Y}^{teU} = (Y^{new(U)} - \mathbf{b}^{teU})./\mathbf{w}^{teU} = (Y^{new(U)} - R^{teU}\mathbf{h})./R^{teU}\mathbf{g}$,

where $R^{teU} = K_{X^{teU}X^{teL}}(L^{teL} + \lambda I)^{-1}$.

With the use of $\mathbf{W} = R\mathbf{G}, \mathbf{B} = R\mathbf{H}, \mathbf{w} = R\mathbf{g}, \mathbf{b} = R\mathbf{h}$, we allow more flexible transformations than mean-centering and variance-scaling while assuming that the transformations are smooth w.r.t $X$. We will illustrate the advantage of the proposed method in the experimental section.

### 3.3 A Kernel Mean Embedding Point of View

After the transformation from $X^{teL}$ to $X^{new(L)}$, we build a Gaussian Process on $(X^{tr}, Y^{tr})$ and predict on $X^{new(L)}$ to get $Y^{new(L)}$. This is equivalent to estimating $\hat{\mu}[P_{Y^{new(L)}}]$ using conditional distribution embeddings [11] with a linear kernel on $Y$: $\hat{\mu}[P_{Y^{new(L)}}] = \hat{\mathcal{U}}[P_{Y^{tr}|X^{tr}}]\hat{\mu}[P_{X^{new(L)}}] = \psi(\mathbf{y}^{tr})(\phi(\mathbf{x}^{tr})^\top \phi(\mathbf{x}^{tr}) + \lambda I)^{-1}\phi^\top(\mathbf{x}^{tr})\phi(\mathbf{x}^{new(L)}) = (K_{X^{new(L)}X^{tr}}(K_{X^{tr}X^{tr}} + \lambda I)^{-1}Y^{tr})^\top$. Finally we want to find the optimal $\mathbf{G}, \mathbf{H}, \mathbf{g}, \mathbf{h}$ such that the distributions on $Y$ are matched across domains, i.e., $P_{Y^*} = P_{Y^{new(L)}}$. The objective function Eq. 2 is effectively minimizing the maximum mean discrepancy: $||\hat{\mu}[P_{Y^*}] - \hat{\mu}[P_{Y^{new(L)}}]||^2 = ||\hat{\mu}[P_{Y^*}] - \hat{\mathcal{U}}[P_{Y^{tr}|X^{tr}}]\hat{\mu}[P_{X^{new(L)}}]||^2$, with a Gaussian kernel on $X$ and a linear kernel on $Y$.

The transformation $\{\mathbf{W}, \mathbf{B}, \mathbf{w}, \mathbf{b}\}$ are smooth w.r.t $X$. Take $\mathbf{w}$ for example, $\hat{\mu}[P_{\mathbf{w}}] = \hat{\mathcal{U}}[P_{\mathbf{w}|X^{teL}}]\hat{\mu}[P_{X^{teL}}] = \varphi(\mathbf{g})(\phi^\top(\mathbf{x}^{teL})\phi(\mathbf{x}^{teL}) + \lambda I)^{-1}\phi^\top(\mathbf{x}^{teL})\phi(\mathbf{x}^{teL}) = \varphi(\mathbf{g})(L^{teL} + \lambda I)^{-1}L^{teL} = (R^{teL}\mathbf{g})^\top$.

### 3.4 Active Learning

We consider two active learning goals and apply a myopic selection criteria to each:
(1) Active Learning which reduces total predictive covariance [18, 19]. An optimal myopic selection is achieved by choosing the point which minimizes the trace of the predictive covariance matrix conditioned on that selection.
(2) Active Surveying [20, 21] which uses an estimation objective that is the sum of all the labels in the test set. An optimal myopic selection is achieved by choosing the point which minimizes the sum over all elements of the predictive covariance conditioned on that selection.

Now we derive the predictive covariance of the SMS approach. Note the transformation between $\hat{Y}^{teU}$ and $Y^{new(U)}$ is given by: $\hat{Y}^{teU} = (Y^{new(U)} - \mathbf{b}^{teU})./\mathbf{w}^{teU}$. Hence we have $Cov[\hat{Y}^{teU}] = \text{diag}\{1./\mathbf{w}^{teU}\} \cdot Cov(Y^{new(U)}) \cdot \text{diag}\{1./\mathbf{w}^{teU}\}$.

As for $Y^{new(U)}$, since we build on Gaussian Processes for the prediction from $X^{new(U)}$ to $Y^{new(U)}$, it follows: $Y^{new(U)}|X^{new(U)} \sim \mathcal{N}(\mu, \Sigma)$, where $\mu = K_{X^{new(U)}X^{tr}}(K_{X^{tr}X^{tr}} + \lambda I)^{-1}Y^{tr}$, and $\Sigma = K_{X^{new(U)}X^{new(U)}} - K_{X^{new(U)}X^{tr}}(K_{X^{tr}X^{tr}} + \lambda I)^{-1}K_{X^{tr}X^{new(U)}}$.

Note the transformation between $X^{new(U)}$ and $X^{teU}$ is given by: $X^{new(U)} = \mathbf{W}^{teU} \odot X^{teU} + \mathbf{B}^{teU}$. Integrating over $X^{new(U)}$, i.e., $P(\hat{Y}^{new(U)}|X^{teU}, D) = \int_{X^{new(U)}} P(\hat{Y}^{teU}|X^{new(U)}, D)P(X^{new(U)}|X^{teU})dX^{new(U)}$, with $D = \{X^{tr}, Y^{tr}, X^{teL}, Y^{teL}\}$. Using the empirical form of $P(X^{new(U)}|X^{teU})$ which has probability $1/|X^{teU}|$ for each sample, we get: $Cov[\hat{Y}^{new(U)}|X^{teU}, X^{tr}, Y^{tr}, X^{teL}, Y^{teL}] = \Sigma$. Plugging the covariance of $Y^{new(U)}$ into $Cov[\hat{Y}^{teU}]$ we can get the final predictive covariance:

$$Cov(\hat{Y}^{teU}) = \text{diag}\{1./\mathbf{w}^{teU}\} \cdot \Sigma \cdot \text{diag}\{1./\mathbf{w}^{teU}\} \qquad (3)$$

## 4 Experiments

### 4.1 Synthetic Dataset

#### 4.1.1 Data Description

We generate the synthetic data with (using matlab notation): $X^{tr} = randn(80, 1), Y^{tr} = sin(2X^{tr} + 1) + 0.1 * randn(80, 1); X^{te} = [w * \min(X^{tr}) + b : 0.03 : w * \max(X^{tr})/3 + b], Y^{te} = sin(2(rev_w * X^{te} + rev_b) + 1) + 2$. In words, $X^{tr}$ is drawn from a standard normal distribution, and $Y^{tr}$ is a sine function with Gaussian noise. $X^{te}$ is drawn from a uniform distribution with a

location-scale transform on a subset of $X^{tr}$. $Y^{te}$ is the same sine function plus a constant offset. The synthetic dataset used is with $w = 0.5; b = 5; rev_w = 2; rev_b = -10$, as shown in Fig. 1.

### 4.1.2 Results

We compare the SMS approach with the following approaches:
(1) **Only test x**: prediction using labeled test data only;
(2) **Both x**: prediction using both the training data and labeled test data without transformation;
(3) **Offset**: the offset approach [16];
(4) **DM**: the distribution matching approach [16];
(5) **KMM**: Kernel mean matching [9];
(6) **T/C shift**: Target/Conditional shift [12], code is from `http://people.tuebingen.mpg.de/kzhang/Code-TarS.zip`.
To ensure the fairness of comparison, we apply (3) to (6) using: the **original** data, the **mean-centered** data, and the mean-centered+variance-scaled (**mean-var-centered**) data.

A detailed comparison with different number of labeled test points are shown in Fig. 4, averaged over 10 experiments. The selection of which test points to label is done uniformly at random for each experiment. The parameters are chosen by cross-validation. Since KMM and T/C shift do not utilize the labeled test points, the MSE of these two approaches are constants as shown in the text box. As we can see from the results, our proposed approach performs better than all other approaches.

As an example, the results for transfer learning with 5 labeled test points on the synthetic dataset are shown in Fig. 5. The 5 labeled test points are shown as filled blue circles. First, our proposed model, SMS, can successfully learn both the transformation on $X$ and the transformation on $Y$, thus resulting in almost a perfect fit on unlabeled test points. Using only labeled test points results in a poor fit towards the right part of the function because there are no observed test labels in that part. Using both training and labeled test points results in a similar fit as using the labeled test points only, because the support of training and test domain do not overlap. The offset approach with mean-centered+variance-scaled data, also results in a poor fit because the training model is not true any more. It would have performed well if the variances are similar across domains. The support of the test data we generated, however, only consists of part of the support of the training data and hence simple variance-scaling does not yield a good match on $P(Y|X)$. The distribution matching approach suffers the same problem. The KMM approach, as mentioned before, applies the same conditional model $P(Y|X)$ across domains, hence it does not perform well. The Target/Conditional Shift approach does not perform well either since it does not utilize any of the labeled test points. Its predicted support of $P(Y^{te})$, is constrained in the support of $P(Y^{tr})$, which results in a poor prediction of $Y^{te}$ once there exists an offset between the $Y$'s.

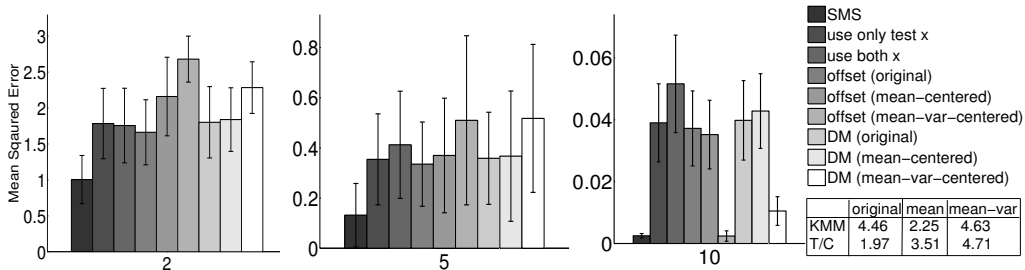

Figure 4: Comparison of MSE on the synthetic dataset with $\{2, 5, 10\}$ labeled test points

## 4.2 Real-world Dataset

### 4.2.1 Data Description

We have two datasets with grape images taken from vineyards and the number of grapes on them as labels, one is riesling (128 labeled images), another is traminette (96 labeled images), as shown in Figure 3. The goal is to transfer the model learned from one kind of grape dataset to another. The total number of grapes for these two datasets are $19, 253$ and $30, 360$, respectively.

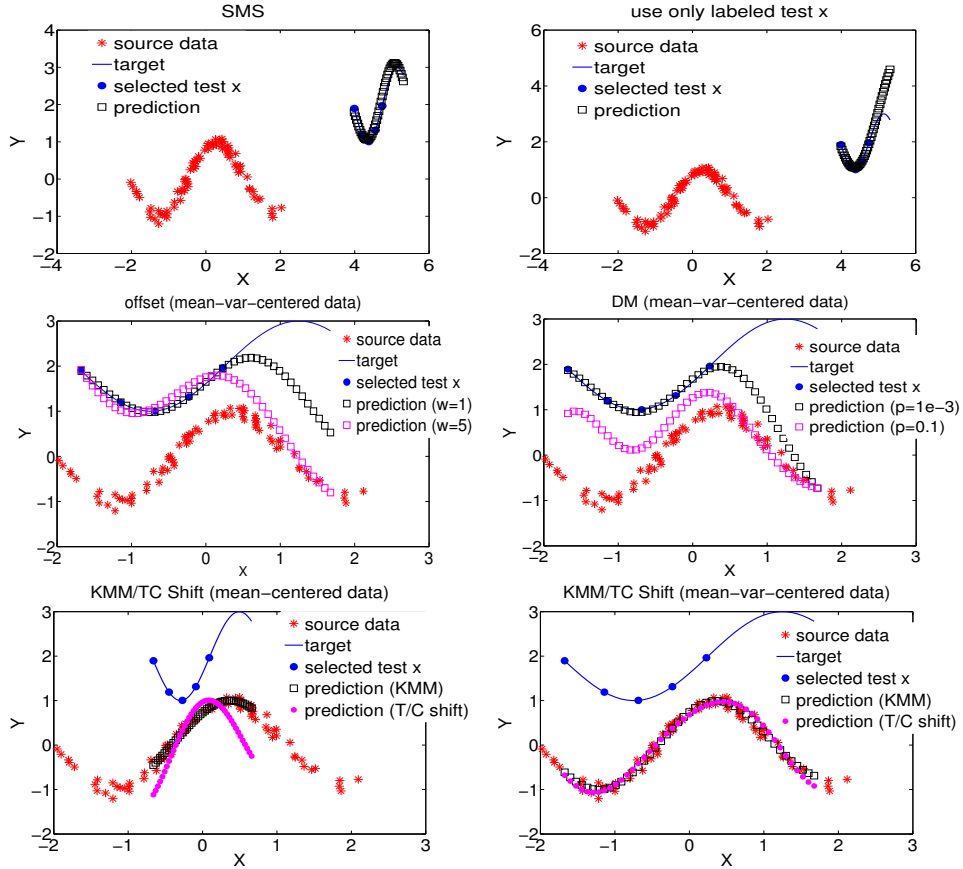

Figure 5: Comparison of results on the synthetic dataset: An example

We extract raw-pixel features from the images, and use Random Kitchen Sinks [1] to get the coefficients as feature vectors [2], resulting in 2177 features. On the traminette dataset we have achieved a cross-validated R-squared correlation of $0.754$. Previously specifically designed image processing methods have achieved an R-squared correlation $0.73$ [3]. This grape-detection method takes lots of manual labeling work and cannot be directly applied across different varieties of grapes (due to difference in size and color). Our proposed approach for transfer learning, however, can be directly used for different varieties of grapes or even different kinds of crops.

### 4.2.2 Results

The results for transfer learning are shown in Table 1. We compare the SMS approach with the same baselines as in the synthetic experiments. For {DM, offset, KMM, T/C shift}, we only show their best results after applying them on the original data, the mean-centered data, and the mean-centered+variance-scaled data. In each row the result in bold indicates the result with the best RMSE. The result with a star mark indicates that the best result is statistically significant at a $p = 0.05$ level with unpaired t-tests. We can see that our proposed algorithm yields better results under most cases, especially when the number of labeled test points is small. This means our proposed algorithm can better utilize the source data and will be particularly useful in the early stage of learning model transfer, when only a small number of labels in the target domain is available/required.

The Active Learning/Active Surveying results are as shown in Fig. 6. We compare the **SMS** approach (covariance matrix in Eq. 3 for test point selection, and SMS for prediction) with:
(1) **combined+SMS**: combined covariance [16] for selection, and SMS for prediction;
(2) **random+SMS**: random selection, and SMS for prediction;
(3) **combined+offset**: the Active Learning/Surveying algorithm proposed in [16], using combined covariance for selection, and the corresponding offset approach for prediction.

From the results we can see that SMS is the best model overall. SMS is better than the Active Learning/Surveying approach proposed in [16] (combined+offset), especially in the Active Surveying result. Moreover, the combined+SMS result is better than combined+offset, which also indicates that the SMS model is better for prediction than the offset approach in [16]. Also, given the better model that SMS has, there is not much difference in which active learning algorithm we use. However, SMS with active selection is better than SMS with random selection, especially in the Active Learning result.

Table 1: RMSE for transfer learning on real data

| # $X^{teL}$ | SMS | DM | Offset | Only test x | Both x | KMM | T/C Shift |
|---|---|---|---|---|---|---|---|
| 5 | **1197±23**\* | 1359±54 | 1303±39 | 1479±69 | 2094±60 | 2127 | 2330 |
| 10 | **1046±35**\* | 1196±59 | 1234±53 | 1323±91 | 1939±41 | 2127 | 2330 |
| 15 | **993±28** | 1055±27 | 1063±30 | 1104±46 | 1916±36 | 2127 | 2330 |
| 20 | **985±13** | 1056±54 | 1024±20 | 1086±74 | 1832±46 | 2127 | 2330 |
| 25 | **982±14** | 1030±29 | 1040 ±27 | 1039±31 | 1839±41 | 2127 | 2330 |
| 30 | 960±19 | **921±29** | 961±30 | 937±29 | 1663±31 | 2127 | 2330 |
| 40 | **890±26** | 898±30 | 938±30 | 901±31 | 1621±34 | 2127 | 2330 |
| 50 | **893±16** | 925±59 | 935±59 | 926±64 | 1558±51 | 2127 | 2330 |
| 70 | 860±40 | 805±38 | 819±40 | **804±37** | 1399±63 | 2127 | 2330 |
| 90 | **791±98** | 838±102 | 863±99 | 838±104 | 1288±117 | 2127 | 2330 |

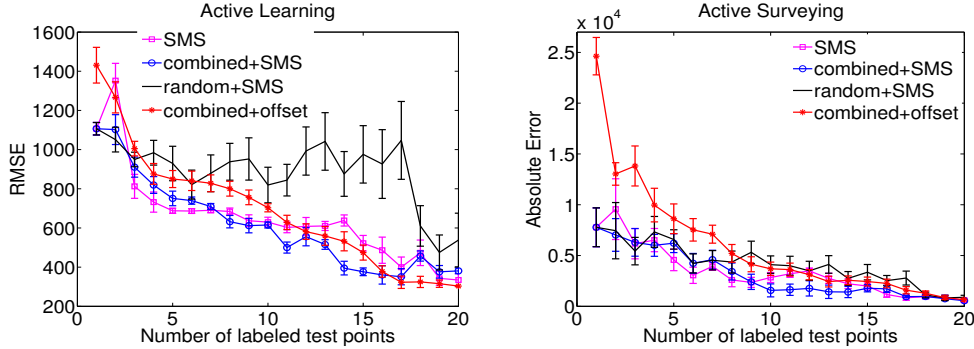

Figure 6: Active Learning/Surveying results on the real dataset (legend: selection+prediction).

## 5 Discussion and Conclusion

Solving objective Eq. 2 is relatively involved. Gradient methods can be a faster alternative but the non-convex property of the objective makes it harder to find the global optimum using gradient methods. In practice we find it is relatively efficient to solve Eq. 2 with proper initializations (like using the ratio of scale on the support for $w$, and the offset between the scaled-means for $b$). In our real-world dataset with 2177 features, it takes about 2.54 minutes on average in a single-threaded MATLAB process on a 3.1 GHz CPU with 8 GB RAM to solve the objective and recover the transformation. As part of the future work we are working on faster ways to solve the proposed objective.

In this paper, we proposed a transfer learning algorithm that handles both support and model shift across domains. The algorithm transforms both $X$ and $Y$ by a location-scale shift across domains, then the labels in these two domains are matched such that both transformations can be learned. Since we allow more flexible transformations than mean-centering and variance-scaling, the proposed method yields better results than traditional methods. Results on both synthetic dataset and real-world dataset show the advantage of our proposed method.

**Acknowledgments**

This work is supported in part by the US Department of Agriculture under grant number 20126702119958.

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
