[Reviews · NeurIPS 2014]

Submitted by Assigned_Reviewer_31

This is an interesting transfer learning paper. It assumes that both the supports and models can shift across domains, and proposed a method to learn both transformations.
Summary: While the approach is shown to outperform both mean-centering and variance-scaling, the performance gain may be due to the assumption that the changes can be modeled by a local-scale shift. If the difference is large such that the change cannot be modeled by a single local-scale shift, does your method run into problems?

Submitted by Assigned_Reviewer_41

This paper proposed a method to allow both the marginal and conditional distributions to change, which is less studied in the literature of transfer learning.

In general, the paper is well written. But one thing is a bit vague for more general audience, i.e., why smoothness should be imposed and is so important in the proposed method. The authors should provide more details in the paper.

This paper is motivated from [16] and [12], which assume the test X and Y can be transformed into new representations where the statistical property is somehow matched with corresponding domains. Though a bit incremental, the proposed method addressed an interesting transfer learning problem, and the experiments are sufficiently done to show the effectiveness of the proposed method.

Minor issues:
1. "mean-var-centered" should be "mean-var-scaled" in Figures 4 and 5.
Summary: It's interesting to see a transfer learning method which allows both the marginal and conditional distributions to change. And the experimental results show good performance of the proposed method. It's a bit incremental (based on [12] and [16]), though.

Submitted by Assigned_Reviewer_42

This paper proposes a transfer learning model. The idea is novel. The presentation needs some clarification. Y is apparently univariate although it is defined as multivariate with dimension dy at the beginning of 3.1. w_te should have the same dimension as W_teL. (1) only regularizes the scale vector over all test data. Why is the location parameter not regularized? Why is R_teL defined in the way proposed? What is the consideration? Any other alternatives? It is quite involved to solve (2). More detail about the algorithm should be provided. Also, what is the computational complexity, especially for high dimensional predictors?
Summary: The transfer learning model proposed in this paper is flexible and novel. However, the presentation needs much improvement. Some detailed argument and explanation should be added to improve clarity.
Author Feedback
Author rebuttal: To reviewer_31:
The non-linearity of the location-scale transformation makes it capable of recovering a fairly wide set of changes, including non-monotonic ones. However, because of the smoothness constraint imposed on the location-scale transformation, it might not recover some extreme cases where the scale or location change is non-smooth/discontinuous. However, under such cases the learning problem by itself would be very challenging.

To reviewer_41:
The smoothness constraint is imposed because we think it is reasonable for real applications and it helps keep the estimation process stable.

The "mean-var-centered" label in the figures is short for "both mean-centered and variance-scaled". Thanks for pointing this out. We will fix the labels.

To reviewer_42:
Yes, in our actual use so far Y is only univariate. The algorithm we proposed easily extends to the multivariate case (Y will become an n by d matrix where d is the dimension of Y) if we allow the transformation $w$ and $b$ on Y to be matrices instead of vectors (just like the transformation we made on X), and change the vector norm in the objective to be a Frobenius norm on the matrix [Y-prediction_on_Y]. We will simplify the description to stick to the univariate case.

w_te has a row-dimension of #all_test_points, and W_teL has a row-dimension of #labeled_test_points. We will clarify the description in the final copy.

Yes, the location parameter could be regularized. We chose not to because in practice the location change could be fairly large. To impose such constraint will result in a choice of very small $\lambda$ before the regularization term and the only effect is to constrain $b$ to not become arbitrarily large. In our experiments, regularizing the scale parameter was sufficient to prevent the optimization of becoming unstable.

R_teL is defined in the way proposed to ensure smoothness. The idea comes from kernel ridge regression where $R=K_{XX}(K_{XX}+\lambda I)^{-1}$ acts as a smoothing matrix on some parameter $\theta$ such that the resulting new parameter $R*\theta$ will be a smooth function w.r.t $X$. A more theoretical explanation is described in the second paragraph of section 3.3. Yes, other ways of ensuring smoothness would be ok.

Yes solving (2) is quite involved. Gradient methods can be a faster alternative but the non-convex property of the objective makes it harder to find the global optimum using gradient methods. Metropolis-Hastings algorithms are usually good for such multi-modal objective. In practice we find it is relatively efficient to solve (2) with proper initializations (like using the ratio of scale on the support for $w$, and the offset between the scaled-means for $b$). For example, in our real-world dataset with 2177-dimension features, it takes about 2.54 minutes on average in a single-threaded MATLAB process on a 3.1 GHz CPU with 8 GB RAM to solve the objective and recover the transformation. As part of the future work we are working on faster ways to solve the proposed objective.